

# Integrating network pharmacology, molecular docking, and molecular dynamics simulations to explore potential compounds and mechanisms of *Coptis chinensis* in treating streptococcal infections

Wanxiang Qi, Bin Shi, Wenqiang Tang, Jiangyong Zeng, Ma Zhuo and Hongcai Ma

Institute of Animal Husbandry and Veterinary Medicine, Xizang Academy of Agriculture and Animal Husbandry Sciences, Lhasa, Xizang, China

Corresponding author
Hongcai Ma, xzmahongcai@126.com

## ABSTRACT

**Background**. *Coptis chinensis*, a prominent herb in traditional Chinese medicine, is widely utilized for its therapeutic effects against Streptococcus infections, though its precise mechanisms of action remain insufficiently understood. This study aims to clarify the potential mechanisms and active compounds of *C. chinensis* in the treatment of Streptococcus.

**Methods**. Active compounds of *C. chinensis* were identified using the Traditional Chinese Medicine Systems Pharmacology Database and Analysis Platform (TCMSP) database, and their potential targets were predicted from multiple public resources. These targets were intersected with streptococcus-related genes to identify overlapping targets, which were then used to construct a protein–protein interaction (PPI) network and screen for key hub genes. To investigate the pharmacological mechanisms, Gene Ontology and KEGG pathway enrichment analyses were performed. Molecular docking was employed to evaluate the binding affinities between active compounds and core target proteins, followed by molecular dynamics simulations and Molecular Mechanics Poisson-Boltzmann Surface Area (MM-PBSA) calculations to assess binding stability and free energy.

**Results**. A total of 24 active compounds were identified, along with 180 overlapping targets related to streptococcal infection. PPI network analysis revealed ten key hub genes, including IL1β, IL6, and MMP9. Enrichment analysis suggested that *C. chinensis* may inhibit the TLR4/NF-κB inflammatory pathway to modulate host immunity and mediate lipid metabolism reprogramming to restrict pathogen proliferation. Several core targets were also enriched in pathways related to extracellular matrix remodeling and immune regulation, indicating potential indirect effects on host–pathogen interface interactions. Molecular docking and simulation confirmed stable binding between major active ingredients and streptococcus-associated proteins.

**Conclusion**. This study provides mechanistic insights into the multi-component, multi-target, and multi-pathway effects of *C. chinensis* against streptococcal infections. The findings offer a theoretical basis for future experimental validation and clinical translation.

## INTRODUCTION

Streptococcus is a genus of Gram-positive cocci, predominantly facultative anaerobes. Based on their hemolytic properties, they can be classified into alpha-hemolytic, beta-hemolytic, and gamma-hemolytic streptococci (*Renzhammer et al., 2020*; *Wong et al., 2022*). Notably, several species, such as *Streptococcus pneumoniae*, *Streptococcus agalactiae*, and *Streptococcus pyogenes*, are common human pathogens. A 2023 survey of 2,274 pneumonia patients in a specific area of Japan revealed that the prevalence of Streptococcus pneumoniae exceeded 20% (*Hamaguchi et al., 2023*). Among these, *Streptococcus pyogenes* stands out as a significant bacterial pathogen, infecting at least 700 million people annually, with a mortality rate ranging from 15% to 30% (*Wong et al., 2022*). *Streptococcus agalactiae* is often isolated from pregnant women and newborns globally, leading to early and late-onset infections in infants, and it has also been implicated in outbreaks affecting various livestock and aquatic species (*Bobadilla et al., 2021*; *Khunrang, Pooljun & Wuthisuthimethavee, 2023*).

In the realm of animal husbandry, streptococci are of critical concern. For instance, *Streptococcus suis* is a major pathogen in pigs, emerging as a significant zoonotic threat worldwide. Research indicates that *Streptococcus suis* can cause meningitis, arthritis, and sepsis in pigs, resulting in substantial economic losses for the swine industry, with infection rates exceeding 20% in intensive farming settings (*Goyette-Desjardins et al., 2014*). Additionally, *Streptococcus dysgalactiae* and *Streptococcus uberis* are primary pathogens responsible for mastitis in dairy cows, with infection rates in *Streptococcus uberis* reaching 30%, severely impacting dairy production and quality (*Bolbol et al., 2017*). In sheep, infections caused by *Streptococcus ovis* exhibit a high incidence rate, with reports indicating rates of up to 15%, leading to severe sepsis and mortality, particularly in pastoral regions of China (*al-Quarawi et al., 1995*).

Although numerous antibiotics are available for treating streptococcal diseases, the misuse of these drugs has led to a yearly increase in antibiotic resistance among streptococcal strains (*Bilgin et al., 2017*). Natural compounds, generally associated with fewer toxic side effects and lower risks of resistance, represent a promising avenue for developing anti-streptococcal therapies (*Yang et al., 2019*). Consequently, researchers are exploring traditional Chinese medicine (TCM) and various natural compounds to identify new "green and environmentally friendly antibacterial agents" for treating streptococcal infections.

*Coptis chinensis* (*C. chinensis*, *Huanglian*) is a traditional Chinese medicinal herb with a bitter taste and cold nature, and it has been used for over two thousand years for its properties in clearing heat, drying dampness, detoxifying, and expelling parasites (*Zhang et al., 2018*). Modern pharmacological studies have shown that *C. chinensis* exhibits significant antibacterial activity against a broad range of Gram-negative and Gram-positive bacteria, including *Shigella* spp., *Salmonella* spp., *Escherichia coli*, and various Streptococcus

species (*Teggi et al., 2019*). It is a common ingredient in several TCM formulations that demonstrate anti-streptococcal effects, and its therapeutic potential against streptococcal infections has been partially validated in prior studies (*Ferguson, Polskaia & Tokuno, 2017*; *Wang et al., 2020*). However, current research on its anti-streptococcal mechanisms remains limited. Most studies focus on *in vitro* antibacterial activity, with insufficient systematic investigation into its bioactive components and pharmacological mechanisms. In addition, molecular-level studies on its targets and signaling pathways are lacking, which hinders its further development as a modern therapeutic agent.

Network pharmacology is an interdisciplinary approach that integrates systems biology, bioinformatics, and pharmacology to explore the complex interactions between drugs and biological systems from a holistic perspective (*Matsushita et al., 2023*). Unlike traditional pharmacological methods that focus on a "one drug–one target" paradigm, network pharmacology is particularly suitable for studying TCM, which is characterized by "multiple components, multiple targets, and multiple pathways" (*Zhang et al., 2019*). Traditional approaches often fail to reveal the synergistic effects of multiple active ingredients, whereas network pharmacology enables the construction of drug–target–disease networks to systematically analyze the comprehensive therapeutic actions of multi-component medicines. This approach not only aids in identifying novel therapeutic targets but also contributes to optimizing treatment strategies, enhancing efficacy, and reducing adverse effects (*Hopkins, 2008*). Complementary to this, molecular docking serves as a theoretical tool to predict binding conformations and affinities between ligands and receptors (*Mansoori et al., 2020*), while molecular dynamics (MD) simulations further explore molecular interactions and stability at the atomic level. The integration of these computational approaches provides strong technical support for elucidating the mechanisms of TCM and has become increasingly vital in modern drug development.

Given the unclear mechanisms underlying the anti-streptococcal effects of *C. chinensis*, this study employs a combined approach of network pharmacology, molecular docking, and molecular dynamics simulation to systematically explore its bioactive compounds, potential targets, and key signaling pathways (Fig. 1). For the first time, we constructed an integrated "*C. chinensis*–target–pathway–streptococcal disease" network model to serve as a preliminary framework for uncovering its molecular mechanisms. This study aims to provide theoretical support for the modernization of TCM and the development of novel therapeutic agents.

## MATERIALS & METHODS

### Screening of effective active ingredients and target prediction in *c. chinensis*

In this study, a multi-database integrative strategy was employed to systematically identify the bioactive compounds of *C. chinensis* and predict their potential targets, ensuring both data comprehensiveness and accuracy. Initially, the TCMSP database (https://www.tcmsp-e.com/tcmsp.php) was used to screen the active compounds of *C. chinensis*, with oral bioavailability (OB) $\geq$ 20% and drug-likeness (DL) $\geq$ 0.1 set as selection criteria to ensure favorable pharmacokinetic properties. To address potential

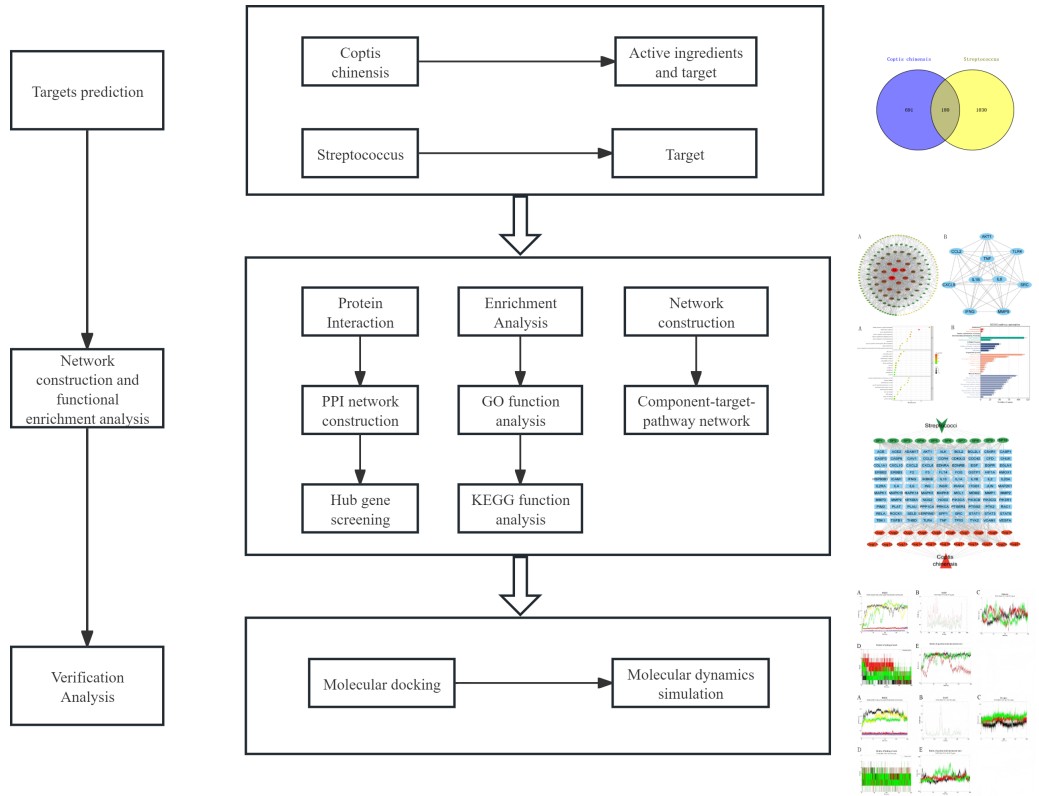

**Figure 1** Analysis flow chart.

limitations of a single database and to increase compound coverage, data were further supplemented with information from the TCMID database and relevant literature.

Target prediction for the selected bioactive compounds was conducted using five databases: tCMSP (old.tcmsp-e.com), DrugBank (go.drugbank.com), SwissTargetPrediction (swisstargetprediction.ch), BATMAN-TCM (ncpsb.org.cn), and TargetNet (targetnet.scbdd.com). Specific thresholds were applied where applicable, including Probability > 0 for SwissTargetPrediction and Score cutoff > 20% with $P < 0.05$ for BATMAN-TCM. These databases incorporate diverse methodologies—including experimental data, computational prediction, machine learning, and network pharmacology—complementing each other to enhance both the breadth and reliability of target identification. To standardize data formats, target names retrieved from TCMSP and TargetNet were converted into official gene symbols using the UniProt database. All predicted targets were subsequently integrated and duplicates removed, resulting in a comprehensive target library for *C. chinensis*.

## Target acquisition and key target intersection in streptococcal disease

To comprehensively collect streptococcal disease–associated targets, four databases were queried: GeneCards (http://www.genecards.org), OMIM (https://omim.org/), TTD (https://db.idrblab.net/ttd/), and DisGeNET (http://www.disgenet.org). These databases

encompass diverse layers of information, including genetic, clinical, and pharmacological associations, thereby providing a multidimensional overview of disease-related genes. After integrating the results and removing duplicate entries, a relatively comprehensive dataset of streptococcal disease–related targets was constructed. The predicted targets of *C. chinensis* bioactive compounds were then intersected with the disease-related targets to identify potential therapeutic targets.

## PPI network construction and hub gene identification

The intersection of predicted drug targets and disease-related targets was input into the STRING database (https://cn.string-db.org) with the species set to Homo sapiens and the minimum required interaction score set to high confidence (>0.7); other parameters were kept at default. After removing disconnected nodes (PPI) network data were exported in TSV format and imported into Cytoscape 3.9.1 for network construction and visualization.

To identify potential key targets, topological features of the network nodes were further analyzed using the CytoNCA plugin in Cytoscape, focusing on three main metrics: Degree, Betweenness Centrality, and Closeness Centrality. Based on a comprehensive evaluation of these metrics, the top 10 hub genes were selected as key candidate targets for subsequent analyses.

## GO and KEGG pathway enrichment analysis

To elucidate the functional characteristics and associated signaling pathways of the potential targets, GO and KEGG enrichment analyses were performed on 180 intersecting genes using the DAVID database (https://davidbioinformatics.nih.gov/). The analyses were conducted through the "Functional Annotation" module, with the species set to *Homo sapiens* and the background gene list corresponding to the human reference genome. GO analysis included annotations in three categories: biological process (BP), cellular component (CC), and molecular function (MF), while KEGG analysis was used to identify relevant signaling and disease pathways. Given the moderate size of the gene set, a significance threshold of $p < 0.05$ without multiple testing correction was applied to retain more biologically meaningful entries for subsequent analyses. GO results were ranked by gene count and the top ten terms of each category (BP, CC, MF) were selected for visualization. All KEGG pathways meeting the criteria were included. Enrichment results were visualized using the Bioinformatics online platform (https://www.bioinformatics.com.cn) through GO bubble plots and KEGG classification bar charts, illustrating the functional distribution and pathway characteristics of the targets.

## Construction of drug ingredient-target-pathway network

The top 10 pathways with the most targets identified in the KEGG pathway enrichment analysis, along with target association information and the corresponding numbers of active ingredients from *C. chinensis*, were imported into Cytoscape 3.9.1 software to construct a drug ingredient-target-pathway network diagram. This network diagram can provide insights into the mechanisms by which *C. chinensis* exerts its therapeutic effects.
## Molecular docking validation

To evaluate the binding potential between candidate active compounds and key target proteins, this study selected 10 critical protein targets, whose three-dimensional structures were primarily obtained from the Research Collaboratory for Structural Bioinformatics Protein Data Bank (RCSB PDB; https://www.rcsb.org/) database. Priority was given to crystal structures with co-crystallized ligands or resolutions better than 2.5 Å. Active sites were defined based on the co-crystallized ligand binding pockets and relevant literature reports, and further refined with grid predictions from AutoDock Tools to ensure accuracy and biological relevance of the docking regions. Molecular docking of 15 potential active compounds was performed using AutoDock 4.2, resulting in a total of 28 protein–ligand complexes. Binding free energy was used as the primary evaluation metric to preliminarily screen the binding affinities of all complexes.

## Molecular dynamics simulation and Molecular Mechanics Poisson-Boltzmann Surface Area binding free energy calculation

Protein–ligand complexes with favorable binding free energies from molecular docking screening were further subjected to molecular dynamics (MD) simulations using GROMACS 2024.2 to evaluate their stability and binding behavior under dynamic physiological conditions. The protein was parameterized with the CHARMM36-jul2021 force field, while ligand parameters were generated using CGenFF. TIP3P water molecules were used as the solvent model. Following energy minimization and equilibration under NVT and NPT ensembles, production MD simulations were conducted for 200 ns at 310 K and 1 atm. Each simulation was independently repeated three times to ensure data reliability. Stability and flexibility of the complexes were assessed by analyzing RMSD, RMSF, hydrogen bond counts, and radius of gyration (Rg) from the simulation trajectories (*Mejia-Gutierrez et al., 2021*). Detailed binding mode analyses were based on conformations from the final stage of MD simulations, focusing on hydrogen bonds, hydrophobic interactions, and $\pi$-$\pi$ stacking between ligands and key residues within the protein binding pockets to elucidate the molecular basis of stable binding. Additionally, binding free energies were calculated using the Molecular Mechanics Poisson-Boltzmann Surface Area (MM-PBSA) method over the 191–200 ns interval, providing a quantitative theoretical basis for binding affinity evaluation.

## RESULTS

### Active ingredients of *c. chinensis* and potential targets for streptococcal disease

From the TCMSP database, 48 active ingredients of *Coptis chinensis* were retrieved. After applying the screening criteria of OB > 20% and DL > 0.1, 24 active ingredients were selected for further analysis. The target genes for these ingredients were identified using five databases: 202 targets from TCMSP, 47 from DrugBank, 621 from SwissTargetPrediction, 101 from BATMAN-TCM, and 260 from TargetNet. After removing duplicates and converting protein names to gene symbols, a total of 871 unique targets were identified.

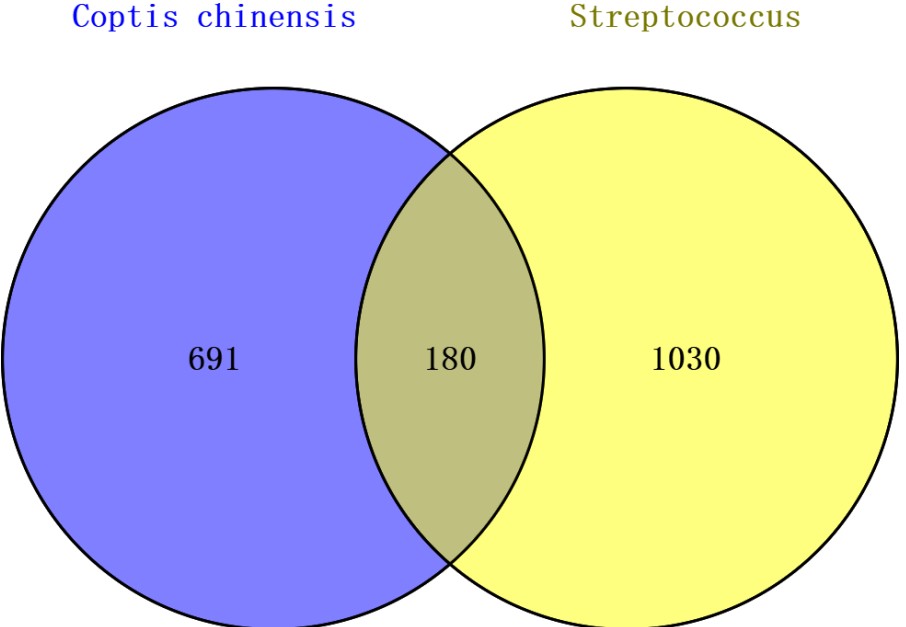

Coptis chinensis          Streptococcus

691          180          1030

Figure 2 **Venn diagram of targets related to *Coptis chinensis* and streptococcal disease.**

For streptococcal disease, 1,198 target genes were gathered from DisGeNET, supplemented by a literature review. Additional relevant targets were retrieved from GeneCards, OMIM, and TTD databases. After eliminating duplicates and converting protein names to gene symbols, a total of 1,210 target genes related to streptococcal disease were identified.

When intersecting the 871 *Coptis chinensis* targets with the 1,210 streptococcal disease targets, 180 common targets were identified, which could potentially be involved in *Coptis chinensis*'s therapeutic effect on streptococcal disease (Fig. 2, Table S1).

## Construction of protein–protein interaction (PPI) network and hub gene screening

A protein–protein interaction (PPI) network was constructed using the STRING database, comprising 179 nodes and 1,546 edges, with an average node degree of 17.2 and an average local clustering coefficient of 0.52 (Fig. 3A). In this network, nodes represent protein targets, and edges indicate potential functional or physical interactions between targets. The size and color intensity of each node are proportional to its degree value, with higher-degree nodes positioned closer to the network core.

Topological properties of the network nodes were systematically analyzed using the CytoNCA plugin. Node importance was comprehensively assessed based on Degree, Betweenness, and Closeness centrality metrics, leading to the identification of 10 key hub genes: TP53, ACTB, AKT1, TNF, HIF1A, IL6, MYC, EGFR, IL1B, and ALB. Among these, TP53 exhibited the highest degree value (125) and was located at the network core,
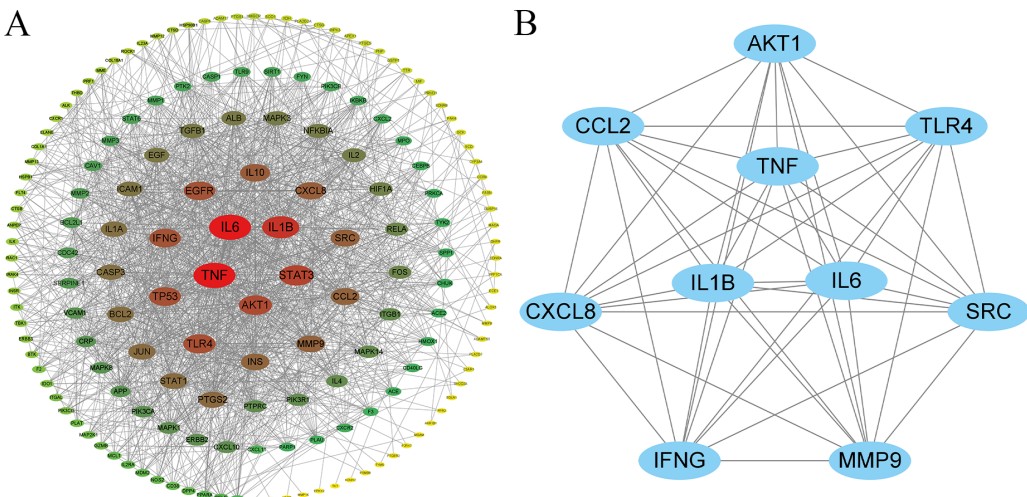

**Figure 3  Key protein–protein interaction (PPI) and hub gene network diagram of *C. chinensis* and streptococcal disease.**

suggesting its potential pivotal regulatory role in the anti-streptococcal mechanism of *C. chinensis* (Fig. 3B, Table S2).

## GO and KEGG pathway enrichment analysis

To further investigate the functional characteristics and biological pathways involving the 180 screened common target genes, GO and KEGG enrichment analyses were performed (Fig. 4, Table S3). GO analysis revealed that these targets were mainly involved in key biological processes (BP) such as immune response, signal transduction, and cellular regulation, including "inflammatory response", "cellular response to lipopolysaccharide", "protein phosphorylation", and "positive regulation of transcription", suggesting their potential roles in modulating host immunity and pathogen-related signaling pathways. In terms of cellular components (CC), targets were enriched in "extracellular space", "plasma membrane", "extracellular exosome", and "cytoplasm", indicating their important roles in signal sensing and transduction. At the molecular function (MF) level, significant enrichment in "protein kinase activity", "enzyme binding", "ATP binding", and "protein binding" was observed, implying that these targets may regulate infection-related processes through kinase-mediated signaling.

KEGG pathway classification analysis revealed that the common targets participate in multiple biological systems, including metabolism, environmental information processing, cellular processes, organismal systems, and human diseases. Among these, environmental information processing was dominated by "signal transduction" pathways (19 pathways); organismal systems showed significant enrichment in "immune system" (20 pathways), endocrine system (15 pathways), and nervous system (five pathways); cellular processes encompassed pathways related to cell community behaviors, cell growth and death, and substance transport. In the disease category, enriched pathways were mainly associated with cancer (26 pathways), viral and bacterial infections (22 pathways), and parasitic infections

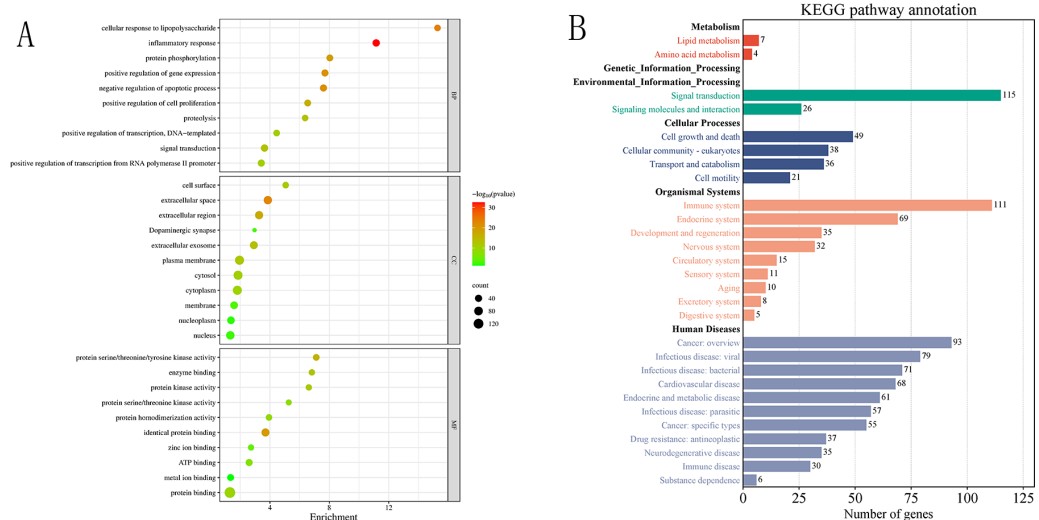

**Figure 4** (A–B) Bubble chart of GO enrichment results and summary chart of KEGG enrichment secondary Classiûcation.

(six pathways), indicating strong enrichment trends. These findings are highly consistent with the GO functional analysis, highlighting the core roles of common targets in immune regulation and signal transduction. Furthermore, many of these pathways include hub targets identified in the protein–protein interaction network analysis (AKT1, MAPK3, TNF), suggesting that these proteins may act as key nodes mediating signal transduction and constitute crucial pathway hubs for *C. chinensis* intervention against streptococcal infection.

## Drug component-target-pathway network analysis

To better visualize the relationships between *C. chinensis* components, target genes, and Streptococcus-related signaling pathways, a drug component-target-disease pathway network was constructed, comprising 132 nodes and 807 edges (Fig. 5). In this network, blue square nodes represent common targets, green oval nodes indicate Streptococcus-associated signaling pathways, and red hexagonal nodes denote the potential active components of *Coptis chinensis*. The network analysis revealed that several active compounds, including quercetin, coptisine (DPEC), epiberberine, and palmatine, were associated with the highest number of targets. These active components may serve as the key bioactive compounds responsible for the therapeutic effects of *C. chinensis* against Streptococcus infections. Through multiple mechanisms, they collectively contribute to the pharmacological activity of *C. chinensis*, highlighting its potential clinical applications.

## Molecular docking and binding energy analysis

The docking results indicated that all complexes exhibited binding free energies lower than −6 kcal/mol, suggesting strong interactions between these active compounds and their target proteins (Table 1). Notably, seven complexes showed binding free energies below −9 kcal/mol, indicating exceptionally high binding affinities. Among them, the Palmidin

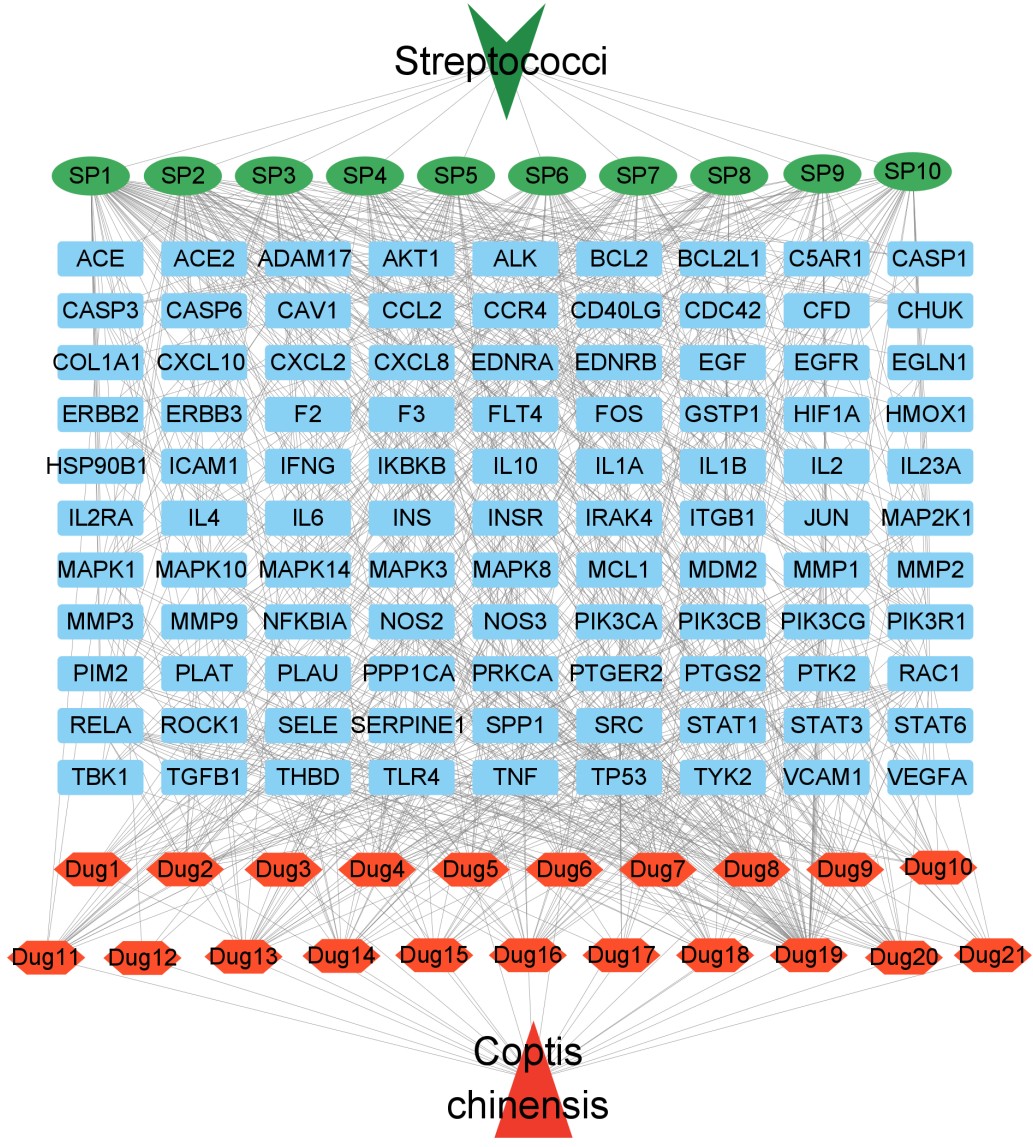

**Figure 5  Network diagram of potential active ingredients, targets, and streptococcal disease signaling pathways in *C. chinensis*.**

A-SRC complex demonstrated the strongest binding affinity, with a binding free energy of −10.7 kcal/mol. Additionally, quercetin, epiberberine, and tetrandrine were identified as key ligand molecules, while SRC, MMP9, and AKT1 emerged as critical target proteins for these active compounds.

## Molecular dynamics simulation and MM-PBSA binding free energy calculation

Palmidin A interacts with the binding pocket of SRC protein but exhibits weak binding affinity and moderate overall stability. As shown in Fig. 6A, the small molecule remains relatively fixed within the binding pocket, while the overall protein conformation undergoes

**Table 1  Molecular docking results of 28 sets of complexes.**

| Small molecular ligand | Receptor protein (PDB ID) | Maximum binding energy (kcal/mol) |
|---|---|---|
| Palmidin A | SRC (1y57) | −10.7 |
| Quercetin | MMP9 (6esm) | −9.8 |
| Quercetin | AKT1 (6s9x) | −9.8 |
| Moupinamide | MMP9 (6esm) | −9.7 |
| Coptisine | SRC (1y57) | −9.2 |
| Tetrandrine | CCL2 (4zk9) | −9.2 |
| Berberine | SRC (1y57) | −9.1 |
| Obacunone | SRC (1y57) | −8.9 |
| Epiberberine | SRC (1y57) | −8.8 |
| Epiberberine | MMP9 (6esm) | −8.6 |
| Obacunoic acid | MMP9 (6esm) | −8.6 |
| Groenlandicine | MMP9 (6esm) | −8.4 |
| Pycnamine | SRC (1y57) | −8.3 |
| Tetrandrine | IFNγ (6e3l) | −8.1 |
| Tetrandrine | AKT1 (6s9x) | −8 |
| Columbamine | SRC (1y57) | −7.8 |
| Quercetin | IFNγ (6e3l) | −7.8 |
| Quercetin | MMP9 (6esm) | −7.8 |
| Palmatine | MMP9 (6esm) | −7.6 |
| Limonin | TNF (2e7a) | −7.4 |
| DPEC | IL6 (1alu) | −7.1 |
| Quercetin | TLR4 (2Z64) | −7.1 |
| Obacunoic acid | TNF (2e7a) | −7 |
| Quercetin | CCL2 (4zk9) | −6.9 |
| Quercetin | IL1B (1t4q) | −6.8 |
| Moupinamide | TNF (2e7a) | −6.4 |
| Columbamine | CXCL8 (1icw) | −6 |
| Quercetin | TLR4 (2Z64) | −6 |

significant changes. Centroid distance analysis indicates that the distance between the small molecule and the active site ranges from 1.75 to 2.75 nm, suggesting a rigid binding mode, whereas the protein backbone RMSD fluctuates considerably (Fig. 6C). Residue RMSF trajectories reveal high flexibility in the binding pocket region, particularly with pronounced fluctuations at the C-terminal (Fig. 6D). Additionally, the hydrogen bond count and radius of gyration remain stable, indicating no conformational collapse in the system (Figs. 6E, 6F). Binding mode analysis (Fig. 6G) suggests that Palmidin A primarily interacts with key residues through hydrogen bonding, hydrophobic interactions, and $\pi-\pi$ stacking, maintaining a certain degree of binding at the end of the simulation. MM-PBSA calculations (191–200 ns) reveal a binding free energy ($\Delta$G) of approximately −24.08 kJ/mol for Palmidin A with SRC protein, corresponding to a binding affinity (Ki) of approximately 6.06 μM, indicating weak binding. Among the interaction components,

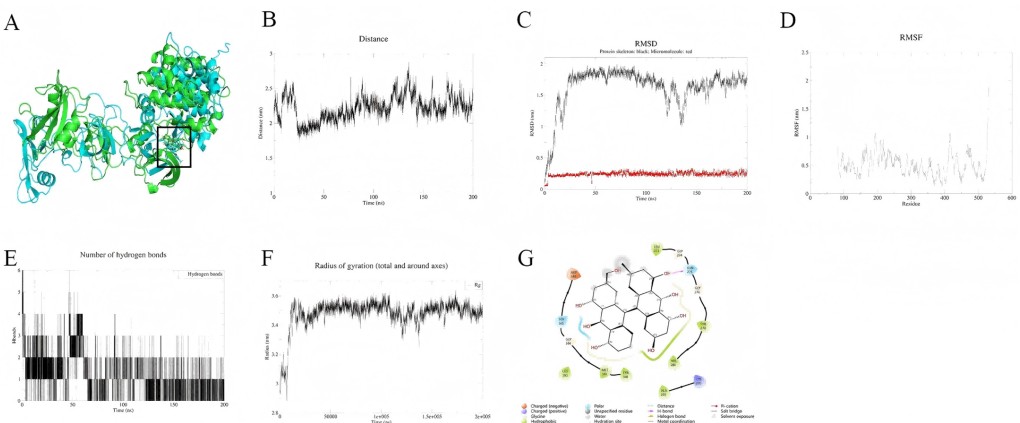

**Figure 6** (A–G) Molecular dynamics simulation results of Palmidin A-SRC.

**Table 2** MM-PBSA binding free energy results.

| Parameters | SRC-Palmidin A | MMP9-Quercetin |
|---|---|---|
| $\Delta G$ (kJ/mol) | $-24.08 \pm 13.6$ | $-98.40 \pm 4.2$ |
| Van der Waals 'contribution | $-135.24 \pm 10.2$ | $-179.61 \pm 12.3$ |
| Electrostatic contribution | $-27.02 \pm 15.3$ | $-44.92 \pm 8.6$ |
| Ki | 61.46 µM | 5.77 nM |

van der Waals interactions ($-135.24$ kJ/mol) contribute the most, whereas electrostatic interactions ($-27.02$ kJ/mol) are relatively minor (Table 2).

In contrast, the MMP9-quercetin complex demonstrates higher stability over the 200 ns simulation. Conformational superposition analysis (Fig. 7A) shows minimal changes in the small molecule within the active pocket, with quercetin dynamically anchored through a hydrophobic core (Phe110, Leu188) and a hydrogen bond network (Tyr219). Its benzene ring forms $\pi-\pi$ stacking with His226 (Fig. 7G). The protein backbone RMSD stabilizes at 0.3–0.4 nm after 30 ns (Fig. 7C), while the RMSD of quercetin remains below 0.1 nm, indicating rapid conformational convergence. Residues at the binding site (Leu181–Tyr223) exhibit low flexibility (Fig. 7D), and the distance between the small molecule and the active site remains stable at 0.3–0.5 nm (Fig. 7B). The hydrogen bond count and radius of gyration show minimal fluctuations (Figs. 7E, 7F), further supporting the tight binding of the complex. MM-PBSA calculations (191–200 ns) indicate that the binding free energy ($\Delta G$) of quercetin with MMP9 is $-107.9$ kJ/mol, with a corresponding binding affinity of $5.77 \times 10^{-9}$ nM, suggesting strong binding. The binding energy is mainly derived from van der Waals forces ($-179.6$ kJ/mol) and electrostatic interactions ($-48.9$ kJ/mol) (Table 2).

To assess the stability of the simulations, three independent simulations were conducted for each complex system (Figs. 8 and 9). Multi-trajectory analysis indicates that the stability of the Palmidin A-SRC complex is independent of initial conditions, with its binding mode

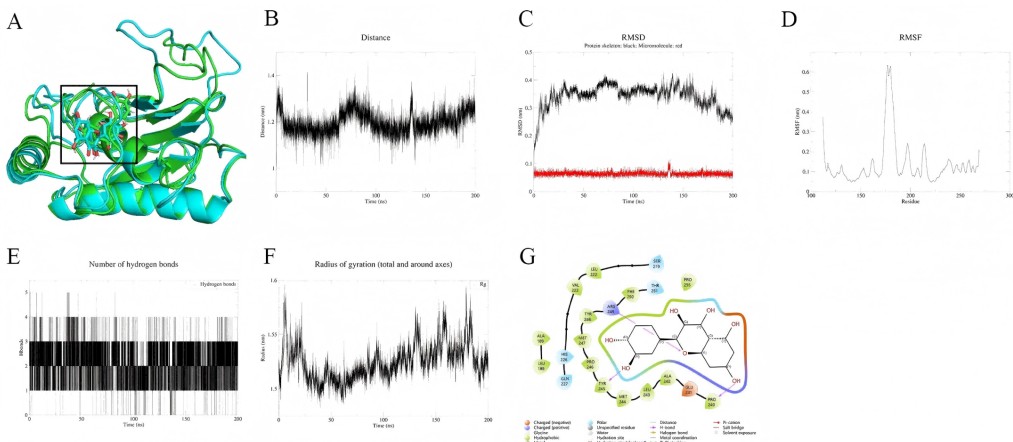

**Figure 7  (A–G) Molecular dynamics simulation results of Quercetin-MMP9 protein.**

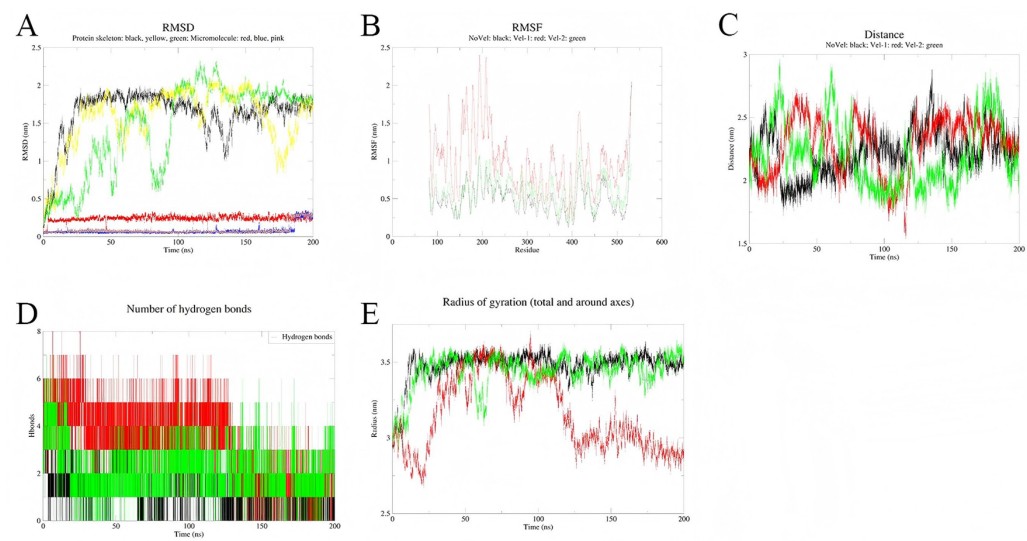

**Figure 8  Integrated analysis chart of multi-trajectory dynamic parameters of Quercetin-MMP9 complex.** (A) RMSD; (B) RMSF; (C) distance between small molecules and protein active sites; (D) Hbnum; (E) gyrate.

achieved through the coordination of dynamic hydrogen bond networks and protein conformational adjustments (Fig. 8A). Meanwhile, the three RMSD curves of the MMP9-quercetin complex nearly overlap (Fig. 9A), with a maximum centroid distance deviation of only 0.3 nm (Fig. 9C). The hydrogen bond network (Fig. 9D) and radius of gyration (Fig. 9E) exhibit consistent dynamic patterns, further confirming the stability of the complex. These findings demonstrate good reproducibility of the simulation data, aligning with the technical standards of molecular dynamics research and ensuring the reliability of the study conclusions (*Shirts et al., 2017*).

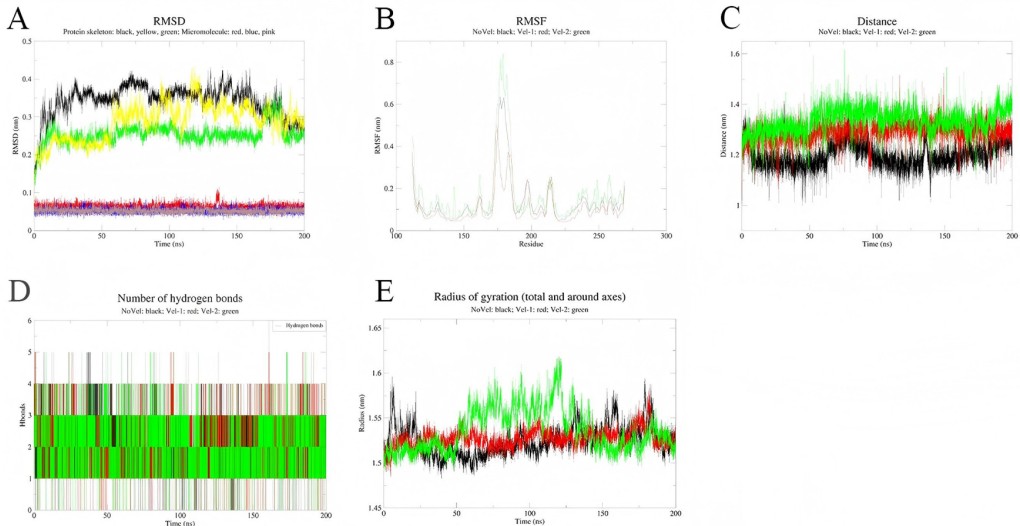

**Figure 9** **Integrated analysis chart of multi-trajectory dynamic parameters of Palmidin A-SRC complex.** (A) RMSD; (B) RMSF; (C) distance between small molecules and protein active sites; (D) Hbnum; (E) gyrate.

# DISCUSSION

The effects and mechanisms of *Rhizoma Coptidis* and its active ingredients in treating various diseases caused by pathogenic streptococci have been widely studied (*Du et al., 2020*; *Li et al., 2018b*). However, the diversity of its active ingredients and pharmacological effects means that traditional research methods focused on a single drug, disease, or target are insufficient to elucidate its mechanisms of action. This study is the first to systematically analyze the potential active ingredients and mechanisms of *Rhizoma Coptidis* in treating streptococcal diseases from the perspectives of multiple components, targets, and pathways. Additionally, molecular docking and kinetic simulation methods were employed to validate the reliability of the analytical results, providing new insights into the complex systems of Rhizoma Coptidis in disease treatment.

Through network pharmacology screening, this study identified 180 potential targets. Among them, compounds such as quercetin, DPEC, and epiberberine emerged as key components of *C. chinensis* for the treatment of streptococcal disease due to their large number of common targets. Quercetin, a flavonoid widely found in plants, exhibits significant anti-inflammatory activity, primarily through the inhibition of inflammatory factors. *Barrientos et al. (2013)* also discovered that quercetin inhibits the growth of streptococci. DPEC, an important active compound in *C. chinensis*, exhibits promising clinical potential in antibacterial, anti-inflammatory, and antioxidant applications (*Gong et al., 2019*). Epiberberine, a representative alkaloid, has demonstrated potent antibacterial and anti-inflammatory properties, making it a promising candidate for various diseases (*Liu, Li & He, 2020*).

Based on the hub gene algorithm of the PPI network, this study identified 10 key host genes, including IL1β, IL6, SRC, and interferon gamma (IFNG), as core targets of *C. chinensis* in the treatment of streptococcal disease. This suggests that *C. chinensis* exerts its therapeutic effects through these targets. The inflammatory response elicited by streptococci is closely linked to mediators such as IL1β and IL6, which streptococci manipulate to evade immune destruction, enhance inflammatory mediators, and increase inflammatory macrophages. This results in a persistent inflammatory environment. IFNG plays a key role in moderating the local inflammatory response during streptococcal infection by regulating pro-inflammatory cytokines in macrophages, aiding bacterial clearance and preventing progression to severe infection, making it a potential specific target for bacterial disease treatmen (*Ivin et al., 2017*). Toll-like receptor 4 (TLR4) is crucial in initiating immune responses in the early stages of streptococcal infection. It recognizes pathogen-associated molecular patterns on the streptococcal surface, triggering signaling pathways that activate immune cells, such as macrophages and dendritic cells, promoting antigen presentation and specific immune responses while initiating inflammatory responses to contain pathogen spread (*Akira, Uematsu & Takeuchi, 2006*; *Medzhitov, 2007*). During the persistent inflammation stage, IL1β, IL6, and tumor necrosis factor (TNF) significantly amplify inflammation. IL1β induces inflammatory mediator production, promotes fever responses, and contributes to tissue damage (*Dinarello, 2009*); IL6 activates immune cells, triggers acute phase reactions, and enhances inflammation (*Tanaka, Narazaki & Kishimoto, 2014*); TNF heightens immune cell bactericidal activity, induces vascular reactions, and can cause tissue damage at high concentrations (*Tracey & Cerami, 1994*). In regulating immune cell function, IFNG and AKT1 are critical. IFNG enhances macrophage function, facilitates antigen presentation, and regulates cellular immunity (*Biron et al., 1999*); AKT1 manages immune cell survival, metabolism, and stress responses, playing a role in immune cell activation (*Manning & Toker, 2017*). Additionally, matrix metalloproteinase 9 (MMP9) may promote bacterial spread by degrading the extracellular matrix, facilitating bacterial dissemination, enhancing inflammatory responses, and potentially delaying tissue repair (*Visse & Nagase, 2003*). Collectively, these proteins interact during streptococcal infection, forming a complex network of immune and inflammatory responses. *C. chinensis* acts on these targets through its effective ingredients, contributing to the treatment of streptococcal disease.

Molecular docking results showed that the primary active ingredients of *C. chinensis* exhibited strong docking activity with core targets, which was further validated through molecular dynamics simulations and MM-PBSA binding free energy calculations. Notably, the binding site of quercetin with MMP9 was the most stable, indicating high intrinsic biological activity. These findings offer insights into the potential mechanisms by which *C. chinensis* treats streptococcal diseases and could serve as a reference for clinical applications.

To gain deeper insights into the mechanisms by which *C. chinensis* (Huanglian) acts against streptococcal infection, this study conducted GO functional and KEGG pathway enrichment analyses on potential therapeutic targets and constructed a drug–target–pathway–disease network. The enrichment results indicated that the anti-streptococcal

effects of Huanglian involve multiple biological processes and signaling pathways. GO analysis revealed significant enrichment of Huanglian targets in biological processes such as "protein phosphorylation", "inflammatory response", and "regulation of cell proliferation", suggesting that Huanglian may modulate host immune signaling by regulating kinase activity to suppress excessive inflammation induced by streptococcal infection. Additionally, the enrichment of target molecules in "cell surface" and "extracellular region" supports the notion that Huanglian's active compounds may exert direct antibacterial effects by interfering with host–pathogen interactions. KEGG pathway analysis further uncovered broad involvement of Huanglian targets in metabolic reprogramming processes, including "lipid metabolism" and "amino acid metabolism", which may restrict streptococcal nutrient uptake and proliferation by altering the host cellular metabolic microenvironment. Notably, enrichment in pathways related to the "immune system" and "bacterial infectious diseases" suggests that Huanglian not only enhances the host innate immune response but may also directly disrupt streptococcal virulence factors, thereby achieving a dual antibacterial effect. Numerous experimental studies have confirmed the significance of these pathways in Streptococcus' growth, apoptosis, and host interactions (*Wang et al., 2017*). In 2017, *Kurosawa et al. (2018)* discovered that the cAMP factor of *Streptococcus pyogenes* promotes adhesion and invasion of pharyngeal epithelial cells *via* the PI3K/Akt signaling pathway. Concurrently, transcriptome analysis of hybrid tilapia infected with *Streptococcus agalactiae* identified the induction of NADPH oxidase and ichthyosidase mediated by the Toll-like receptor pathway as the primary immune response (*Ken et al., 2017*). *Li, Chen & Zhou (2018a)* demonstrated that batryticatine, combined with clindamycin, inhibited severe pneumonia caused by mixed infections of H5N1 influenza virus and *Streptococcus pneumoniae in vitro* and *in vivo via* the NF-κB signaling pathway. These findings suggest that *C. chinensis* may modulate the inflammatory response and immune disorders caused by streptococcal infection through a multi-target, multi-pathway approach, regulating cell signaling pathways to exert its anti-infection effects.

Although this study preliminarily constructed an integrated "*C. chinensis*–target–pathway–streptococcal disease" network model through network pharmacology, molecular docking, and molecular dynamics simulations to reveal the potential active compounds and mechanisms of Huanglian, several limitations remain. First, the current model primarily focuses on host (human) targets and does not systematically incorporate key streptococcal pathogen targets, which may limit its ability to fully capture the direct effects of Huanglian on the pathogen. Second, network pharmacology relies heavily on existing databases and algorithms, and the completeness and accuracy of these data restrict the coverage and predictive power of the model. Moreover, as an association-based approach, it is challenging to directly reflect true causal relationships within the complex *in vivo* physiological environment. Biological factors such as drug absorption, metabolism, and bioavailability also significantly influence actual therapeutic efficacy. Therefore, future research should integrate streptococcal target screening and systematic experimental validation to enhance the biological relevance and clinical translational potential of the

computational model, facilitating the transition from in silico prediction to effective intervention.

## CONCLUSIONS

This study reveals that *C. chinensis* exerts its effects by targeting host core molecules such as IL-1β, IL-6, and MMP9 through active components like quercetin and berberine, demonstrating a multi-component, multi-target, and multi-pathway synergistic effect. The mechanism primarily involves inhibiting the TLR4/NF-κB inflammatory pathway to regulate host immunity and mediating lipid metabolism reprogramming to restrict pathogen proliferation. Furthermore, the enrichment of related targets in extracellular matrix remodeling and immune regulation networks suggests that *C. chinensis* may indirectly inhibit infection by modulating host-pathogen interface functions. This study offers a novel perspective on host-directed antimicrobial mechanisms of traditional Chinese medicine, and in the future, combining pathogen target screening with dual-target intervention strategies will help further elucidate the *C. chinensis*'s potential for host-pathogen synergistic therapy.

## ACKNOWLEDGEMENTS

The authors acknowledge the use of DeepSeek, a generative AI tool, for translation and language polishing during the preparation of this manuscript. The authors carefully reviewed and edited the content after using the tool, and take full responsibility for the final version of the article.

### Funding

The study was supported by the Base and Talent Program of Science and Technology Plan in Tibet Autonomous Region (XZ202401JD0012); The National Key Research and Development Program of China (2022YFD1302101); The central government guides local science and technology development fund projects (XZ202301YD0015C). The funders had no role in study design, data collection and analysis, decision to publish, or preparation of the manuscript.

### Grant Disclosures

The following grant information was disclosed by the authors:
Base and Talent Program of Science and Technology Plan in Tibet Autonomous Region: XZ202401JD0012.
The National Key Research and Development Program of China: 2022YFD1302101.
The central government guides local science and technology development fund projects: XZ202301YD0015C.

### Competing Interests

The authors declare there are no competing interests.

## Author Contributions

- Wanxiang Qi conceived and designed the experiments, performed the experiments, analyzed the data, prepared figures and/or tables, and approved the final draft.
- Bin Shi analyzed the data, prepared figures and/or tables, authored or reviewed drafts of the article, and approved the final draft.
- Wenqiang Tang conceived and designed the experiments, analyzed the data, prepared figures and/or tables, authored or reviewed drafts of the article, and approved the final draft.
- Jiangyong Zeng analyzed the data, authored or reviewed drafts of the article, and approved the final draft.
- Ma Zhuo analyzed the data, authored or reviewed drafts of the article, and approved the final draft.
- Hongcai Ma conceived and designed the experiments, performed the experiments, analyzed the data, prepared figures and/or tables, and approved the final draft.

## Data Availability

Data is available at Figshare:

Ma, Hong-Cai (2025). raw data. figshare. Dataset. https://doi.org/10.6084/m9.figshare.28660550.v1.

Data is also available in the Supplemental Files.

## Supplemental Information

Supplemental information for this article can be found online at http://dx.doi.org/10.7717/peerj.19960#supplemental-information.

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
