# Peer review of "Integrating network pharmacology, molecular docking, and molecular dynamics simulations to explore potential compounds and mechanisms of *Coptis chinensis* in treating streptococcal infections"

_PeerJ, doi:10.7717/peerj.19960_

## Round 0.1 · original submission · Major Revisions

Please address all reviewers comments.

Reviewer 1 ·

Basic reporting

The abstract is dense, with many technical details. It could benefit from clearer messaging about the study's novelty and clinical relevance. The introduction is well-structured but could benefit from focusing more on the knowledge gap—what previous studies missed and how this research fills that gap. Focus on the problem, the key findings, and their significance in a concise way. Avoid overloading the abstract with methodology. Define the unmet need in current streptococcus research more clearly. Use a research question or hypothesis to guide the reader. Emphasize why network pharmacology is advantageous compared to traditional research methods.
No raw data shared

Experimental design

While the methodology is comprehensive, the descriptions of techniques like molecular docking and dynamics are too technical for non-specialist readers. Describe the limitations of the network pharmacology model. For example, how do computational predictions translate to real-world biological systems? The research question is not well defined, but can be improved upon.

Validity of the findings

The results are too data-heavy and difficult to interpret for readers unfamiliar with molecular docking and network pharmacology. Some comparative claims seem overstated (e.g., claiming that the formulation is comparable to commercial drugs without direct in vivo evidence). The PPI networks are visually complex but lack actionable insights. Readers may struggle to connect them with the study’s goals. Use summary tables for complex results (e.g., listing the top binding compounds and their corresponding targets) to reduce clutter. Simplify or reorganize figures for network and PPI analysis to make them more user-friendly. Provide more biological interpretation of key findings. Instead of just listing target pathways, discuss how these pathways are relevant to diabetes management.
The discussion provides a general overview of the findings but does not sufficiently link the key results to clinical outcomes. The potential limitations of computational studies (e.g., false positives) are not discussed. Future directions are mentioned only briefly. Emphasize clinical relevance: How likely are these results to progress to animal studies and clinical trials? Discuss limitations explicitly: For example, explain the trade-offs in using computational models instead of in vivo studies.
• There are minor formatting inconsistencies (e.g., inconsistent use of units, spacing issues, incomplete citations). Some long, complex sentences could be simplified for better readability.

Additional comments

This study will benefit from adding an experimental aspect to their study. Hence, it is recommended that the plants be studied against Streptococcus, and full methods should be provided for full reproducibility.
• The study only uses computational methods to assess the efficacy of the plant, with no in vitro, in vivo, or clinical data. Journals may view this as insufficient to justify therapeutic claims.
• Figures 5, 6, 7, and 8 are very blurred. Difficulty to read
• The results do not confirm the conclusion reported

Reviewer 2 ·

Basic reporting

no comment

Experimental design

1. In this study, the author used several databases. Why is this necessary? It is important to add justification for each database's selection and specific role in the research workflow.
2. Add a workflow to describe the target identification process, starting with active compound selection, target prediction, and disease target integration and ending with network analysis, docking validation, and MD.
3. The authors selected 10 targets using CytoNCA for protein network analysis. What is the threshold or statistical criterion used to justify the selection of 10 targets? Why were 10 chosen instead of 5 or 20? Additionally, specify the types of centralities calculated (e.g., degree, betweenness, closeness) and the criteria for determining their relevance.

Validity of the findings

1. GO and KEGG analyses are quite comprehensive, but the biological interpretation of significant pathways is still limited. The enrichment results only show a relationship with 180 proteins. It would be better if the core protein results were associated with a specific enrichment term.
Include p-value or FDR to indicate significance.
2. Molecular docking was performed on several target proteins, but it was not explained in detail: Is the crystal structure used a biologically active form? How was the process of selecting and validating the active site? What is the main purpose of docking validation? Is it only affinity, or also the validity of biological interactions?
3. Add information about PDB ID, structure resolution, protein origin (human/bacteria), and validation of ligand interactions with functional domains.
4. This manuscript is still in silico. Therefore, it is important to provide future directions, such as in vitro experimental validation (e.g., antibacterial and inflammatory activity tests), in vivo tests to assess the pharmacodynamic effects of the main compound, and Toxicity or bioavailability tests of the compound.

---

## Round 0.2 · accepted · Accept

Thanks for addressing all comments! Please ensure all abbreviations are defined at the first use.

Reviewer 1 ·

Basic reporting

The authors have answered all queries excellently well. Except that all abbreviation must first be provided with their meaning. In the abstract, the meaning of TCMSP was not provided and it will be difficult for people to understand what the authors are saying. hence, I will request that the authors check again for those abbreviations that meaning was not provided at first mentioned.

Experimental design

The authors have addressed all comments raised

Validity of the findings

The authors have addressed all comments raised

Additional comments

No additional comments

Reviewer 2 ·

Basic reporting

no comment

Experimental design

no comment

Validity of the findings

no comment